# Chemical-genetic profiling reveals limited cross-resistance between antimicrobial peptides with different modes of action

Bálint Kintses[1,2,3,12]*, Pramod K. Jangir[1,4,12], Gergely Fekete[1,5,12], Mónika Számel[1,4], Orsolya Méhi[1], Réka Spohn[1], Lejla Daruka[1,4], Ana Martins[1], Ali Hosseinnia[6], Alla Gagarinova[7], Sunyoung Kim[6], Sadhna Phanse[6], Bálint Csörgő[1,10], Ádám Györkei[1,5], Eszter Ari[1,5,8], Viktória Lázár[1,11], István Nagy[9], Mohan Babu[6], Csaba Pál [1]* & Balázs Papp [1,5]*

Antimicrobial peptides (AMPs) are key effectors of the innate immune system and promising therapeutic agents. Yet, knowledge on how to design AMPs with minimal cross-resistance to human host-defense peptides remains limited. Here, we systematically assess the resistance determinants of *Escherichia coli* against 15 different AMPs using chemical-genetics and compare to the cross-resistance spectra of laboratory-evolved AMP-resistant strains. Although generalizations about AMP resistance are common in the literature, we find that AMPs with different physicochemical properties and cellular targets vary considerably in their resistance determinants. As a consequence, cross-resistance is prevalent only between AMPs with similar modes of action. Finally, our screen reveals several genes that shape susceptibility to membrane- and intracellular-targeting AMPs in an antagonistic manner. We anticipate that chemical-genetic approaches could inform future efforts to minimize cross-resistance between therapeutic and human host AMPs.

[1] Synthetic and Systems Biology Unit, Institute of Biochemistry, Biological Research Centre, Szeged, Hungary. [2] HCEMM-BRC Translational Microbiology Lab, Szeged, Hungary. [3] Department of Biochemistry and Molecular Biology, University of Szeged, Szeged, Hungary. [4] Doctoral School of Biology, Faculty of Science and Informatics, University of Szeged, Szeged, Hungary. [5] HCEMM-BRC Metabolic Systems Biology Lab, Szeged, Hungary. [6] Department of Biochemistry, University of Regina, Regina, Saskatchewan, Canada. [7] Department of Biochemistry, University of Saskatchewan, Saskatoon, Saskatchewan, Canada. [8] Department of Genetics, Eötvös Loránd University, Budapest, Hungary. [9] Sequencing Platform, Institute of Biochemistry, Biological Research Centre, Szeged, Hungary. [10] Present address: Department of Microbiology and Immunology, University of California, San Francisco, USA. [11] Present address: Faculty of Biology, Technion – Israel Institute of Technology, Haifa, Israel. [12] These authors contributed equally: Bálint Kintses, Pramod K. Jangir, Gergely Fekete. *email: kintses.balint@brc.hu; cpal@brc.hu; pappb@brc.hu

Antimicrobial peptides (AMPs) play a crucial role in general defense mechanisms against microbial pathogens in all classes of life. Although there is a considerable diversity in their amino acid content, length and structure, AMPs are typically positively charged and amphipathic molecules[1,2]. These properties allow them to adsorb onto the bacterial cell surface and penetrate through the membrane to exert their diverse antibacterial actions[3]. As AMPs have a broad spectrum of activity, considerable efforts have been allocated to the research and development of novel anti-infective compounds originating from AMPs[4,5]. However, the clinical development of AMP therapies, has also raised concerns that these approaches may drive bacterial evolution of resistance to human host-defense peptides[6,7]. As well, therapeutic AMPs are required to be active against pathogenic bacteria, many of which have already evolved resistance against human host AMPs[8]. Therefore, ideally, resistance mechanisms against therapeutic and host AMPs should not overlap.

Accumulating evidence suggests that AMPs differ considerably in their modes of action, which may influence the specific microbial resistance mechanisms against them[1,9]. First, there are substantial differences in the electrostatic interactions and transport processes that lead to the cellular uptake of AMPs[3]. Second, the cellular targets of AMPs are also diverse in nature. For instance, apart from their membrane-disruptive activities, AMPs inhibit intracellular processes such as bacterial DNA and RNA synthesis, translation, cell wall synthesis, and diverse metabolic pathways[1]. However, the extent to which the genetic determinants of resistance differ across AMPs remains unclear, because most of our knowledge comes from case studies characterizing only a limited number of membrane-targeting AMPs[9] (for a list of previously reported resistance genes, see Supplementary Data 1). Therefore, there is an urgent need to comprehensively map the relationships between the modes of action of AMPs and the genetic determinants influencing bacterial susceptibility to them. Understanding these complex relationships would help to rationally choose AMPs for clinical development, which are dissimilar to human host peptides in terms of the underlying resistance mechanisms.

Chemical-genetic profiling is a reverse genetic approach that quantifies the susceptibility of a genome-wide collection of mutant libraries to a set of chemical compounds[10]. By modulating gene dosage (i.e., either by depletion or overexpression), several studies demonstrated the effectiveness of this tool to map cellular targets and genetic determinants of resistance for antibiotics[11–16]. Moreover, antibiotics with similar chemical-genetic interaction profiles, i.e., those with a large overlap between the gene sets influencing resistance to them, are likely to share cellular targets and mechanisms of action[14]. Consistent with this notion, chemical-genetic interaction profiles have been shown to carry information on cross-resistance, i.e., whether resistance evolution to an antibiotic would lead to decreased sensitivity to another antibiotic[17].

Here, we employ a genome-wide chemical-genetic approach to explore the diversity of resistance determinants across AMPs in the model bacterium *Escherichia coli* (*E. coli*). First, we generate a comprehensive chemical-genetic map by measuring how overexpressing each of the ~4400 *E. coli* genes influences the bacterium's susceptibility against 15 AMPs. The set of 15 AMPs are structurally and chemically diverse and include AMPs with well-characterized modes of action, clinical relevance, or crucial role in the human immune defense (Table 1). The resulting chemical-genetic interaction profiles cluster the AMPs according to their modes of action and reveal distinct and often antagonistic resistance determinants against membrane-targeting and intracellular-targeting AMPs. We confirm these results with a complementary chemical-genetic approach by testing the growth effect of a smaller set of 4 selected AMPs against an array of 279 partially depleted essential genes (i.e., hypomorphs)[18,19]. Finally, we analyze the cross-resistance patterns of *E. coli* lines that evolved resistance to AMPs in a recent laboratory evolution study[20]. This analysis confirms that intracellular-targeting AMPs are less likely to induce cross-resistance to membrane-targeting human AMPs than those that share the same broad modes of action.

## Results

**Chemical-genetics reveals AMP resistance-modulating gene sets.** We generated chemical-genetic interaction profiles for a diverse set of AMPs (Table 1) by screening them against a comprehensive library of gene overexpressions in *E. coli*[21].

**Table 1 List and characteristics of AMPs used in this study. Their abbreviation, described mode of action, and clinical relevance (for details see Supplementary Data 7).**

| Name of AMP | Abbreviation | Mode of action | Clinical relevance |
|---|---|---|---|
| Apidaecin IB | AP | Inhibits protein biosynthesis by targeting ribosomes; Interacts with DnaK, GroEL/GroES, FtsH | Yes |
| Bactenecin 5 | BAC5 | Inhibits protein and RNA synthesis | n.a. |
| CAP18 | CAP18 | Disrupts cell membrane | Yes |
| Cecropin P1 | CP1 | Disrupts cell membrane | n.a. |
| Human beta-defensin-3 | HBD-3 | Disrupts cell membrane; Inhibits lipid II in peptidoglycan biosynthesis | n.a. |
| Indolicidin | IND | Inhibits DNA and protein synthesis; Disrupts cell membrane; Inhibits septum formation | Yes |
| LL-37 human cathelicidin | LL37 | Disrupts cell membrane; Induces ROS formation | Yes |
| Peptide glycine-leucine amide | PGLA | Disrupts cell membrane | n.a. |
| Pexiganan | PEX | Disrupts cell membrane | Yes |
| Pleurocidin | PLEU | Disrupts cell membrane; Induces ROS formation; Inhibits protein and DNA synthesis | n.a. |
| Polymyxin B | PXB | Disrupts cell membrane; Induces ROS formation | Yes |
| PR-39 | PR39 | Inhibits protein and DNA synthesis | n.a. |
| Protamine | PROA | Affects cellular respiration and glycolysis; Disrupts cell envelop | n.a. |
| R8 | R8 | n.a. | n.a. |
| Tachyplesin II | TPII | Disrupts cell membrane | n.a. |

*n.a.* no data available

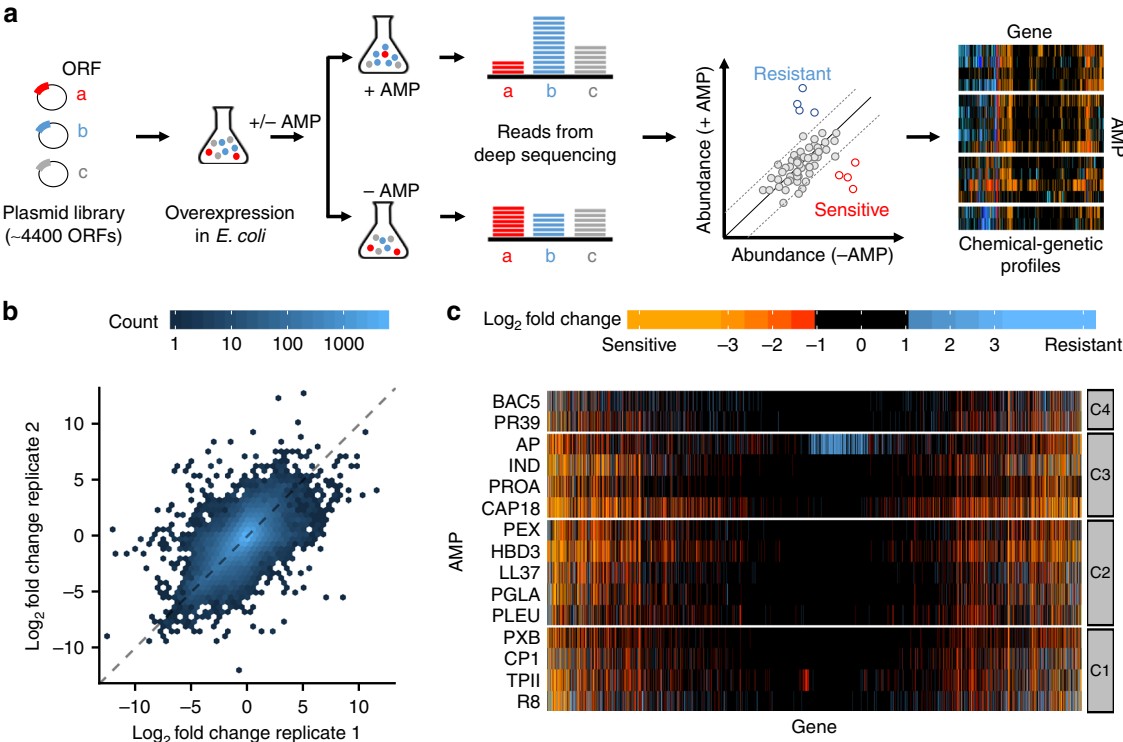

**Fig. 1 Chemical-genetic profiling of AMPs. a** Schematic representation of the chemical-genetic pipeline. The chemical-genetic interactions of ~4400 single gene-overexpressions and 15 different AMPs were measured using a pooled fitness assay with a deep sequencing readout (see Methods). **b** A density scatter plot showing the overall correlation of replicate measurements of the chemical-genetic scores (log₂ fold-change in the relative abundance of each gene in the presence vs absence of each AMP) across all genes and AMPs ($r = 0.63$ and $P = 2.2 \times 10^{-16}$, Pearson's correlation, $n = 53,292$). **c** Heatmap showing the chemical-genetic interaction scores. Resistance-enhancing and sensitivity-enhancing chemical-genetic scores are represented by blue and red, respectively. Groups C1–C4 refer to clusters defined in Fig. 2. Source data are provided as Supplementary Data 2.

Increasing gene dosage is a widely applied approach to reveal the targets of small-molecule antibiotics[22,23]. It also informs on the 'latent resistome', that is, the collection of genes where a change from native expression level enhances resistance to a particular drug[24]. We applied a sensitive competition assay by monitoring growth of a pooled plasmid library, overexpressing all the *E. coli* ORFs (Fig. 1a), as we reported earlier[25]. Specifically, *E. coli* cells carrying the pooled plasmid collection were grown in the presence or absence of one of the 15 AMPs tested, at a sub-inhibitory concentration that increased the doubling time of the whole population by 2-fold. Following 12 generations of growth, the plasmid pool was isolated from each selection and the relative abundance of each plasmid was determined by a deep sequencing readout (see Methods). By comparing plasmid abundances in the presence and absence of each AMP, we calculated a chemical-genetic interaction score (fold-change value) for each gene and identified genes that significantly increase sensitivity (sensitivity-enhancing genes) or decrease sensitivity (resistance-enhancing genes) upon overexpression (Fig. 1a, Supplementary Data 2, see Methods).

To validate our workflow, we took three distinct approaches. First, we tested the reproducibility of the chemical-genetic interaction profiles by correlating the chemical-genetic interaction scores between replicate measurements. The overall correlation was comparable to what has been achieved with arrayed mutants on high-density agar plates[14,26] ($r = 0.63$ from Pearson's correlation, Fig. 1b). This indicates that we measured the growth effects with sufficiently high confidence. Second, we picked 19 overexpression plasmids that showed diverse chemical-genetic interaction scores with multiple AMPs in our screen but did not influence the growth rate of *E. coli* in the absence of AMPs (see

Methods) and performed minimum inhibitory concentration (MIC) measurements on them. Although mutations that affect growth at sub-inhibitory drug dosage do not necessarily alter MIC, we detected a change in MIC in the expected direction for 83% of the tested chemical-genetic interactions (Supplementary Fig. 1). On average, the change in MIC was ~1.6-fold and ~0.7-fold for resistance-enhancing and sensitivity-enhancing gene overexpressions, respectively. Third, we collected examples from the literature where overexpression of an *E. coli* gene has been shown to influence sensitivity to a specific AMP. Despite differences in the used strains and protocols, 69% (9 out of 13) of the literature-curated interactions were captured by our screen (Supplementary Table 1). Taken together, these analyses indicate that our workflow is suitable to measure chemical-genetic interactions between AMPs and gene overexpressions.

**Chemical-genetics groups AMPs with similar features**. We next explored how similarity in the chemical-genetic interaction profiles can inform on the functional and physicochemical similarities of AMPs. To do so, we compiled literature data on known modes of action (Table 1) and computed physicochemical properties for each AMP (see Methods and Supplementary Data 3). Next, we grouped AMPs with similar chemical-genetic interaction profiles using a robust clustering method (see Methods). This procedure resulted in four main clusters, referred to as C1–C4 (Figs. 1c and 2a).

We found that clusters C1 and C2 contain mostly AMPs that target primarily the bacterial membranes, whereas most AMPs in clusters C3 and C4 have intracellular targets (Fig. 2a and Table 1). Membrane-targeting AMPs (C1 and C2) have unique

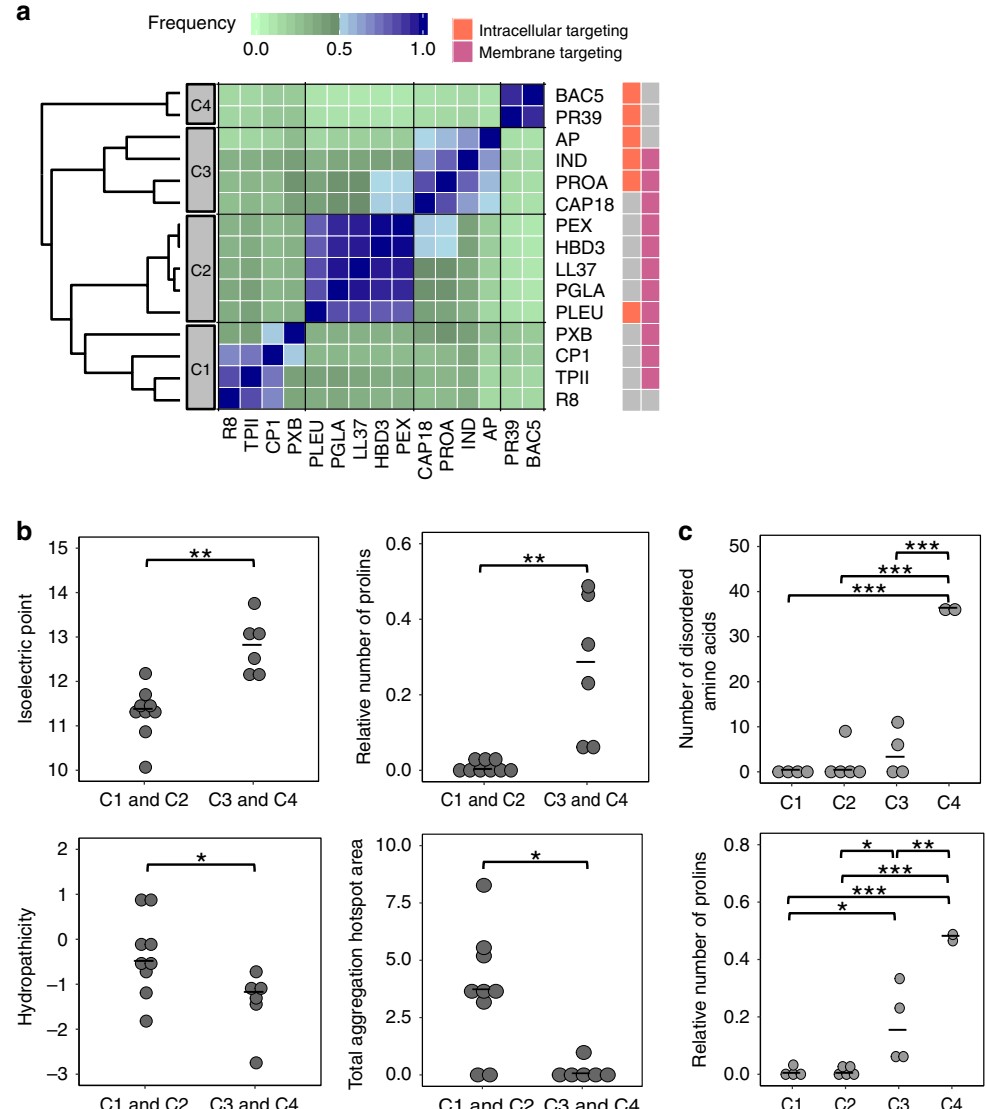

**Fig. 2 Chemical-genetic profiling discriminates membrane-targeting and intracellular-targeting AMPs. a** Heatmap showing the ensemble clustering of the AMPs based on their chemical-genetic profiles (see Methods). For each AMP pair, the color code represents the frequency of being closest neighbors across the ensemble of clusters ($n = 75,000$ clustering). The four major clusters are labeled as C1, C2, C3, and C4. Membrane-targeting and intracellular-targeting broad modes of action are labeled with pink and orange, respectively, on the rightmost side of the figure. Gray color indicates that the specific broad mode of action has not been described or not tested (see Table 1). References describing these activities are provided in Supplementary Data 7.
**b** Most important physicochemical properties that differentiated AMPs in cluster C1, C2 from AMPs in cluster C3, C4. Significant differences: **$P = 0.0026$ and 0.0012 for isoelectric point and relative number of prolines, respectively, * $P = 0.0391$ and $P = 0.0154$ for hydropathicity and total aggregation hotspot area, respectively, two-sided Mann–Whitney U test, $n = 9$ and $n = 6$ for C1, C2 and C3, C4, respectively. **c**, Physicochemical properties that distinguished the clusters when the 4 main AMP clusters were considered separately ($p < 0.05$ ANOVA, Tukey post-hoc test, $n = 15$). Significant differences: ***$P = 1.1 \times 10^{-6}$, $P = 1.3 \times 10^{-6}$ and $P = 4 \times 10^{-6}$ for C1 vs C4, C2 vs C4 and C3 vs C4, respectively in the case of number of disordered amino acids. *$P = 0.034$ and $P = 0.027$ for C1 vs C3 and C2 vs C3, respectively. **$P = 0.0022$ for C3 vs C4. ***$P = 5.5 \times 10^{-5}$ and $P = 4.2 \times 10^{-5}$ for C1 vs C4 and C2 vs C4, respectively, in the case of relative number of prolines. Central horizontal lines represent median values. Source data are provided as Supplementary Data 3.

physicochemical properties (Supplementary Fig. 2). Specifically, they have a lower isoelectric point and proline content, and are substantially more hydrophobic with higher propensity to form secondary structures than C3 and C4 peptides (Fig. 2b). These properties facilitate efficient integration of AMPs into the bacterial membrane where they create pores[27,28]. Notably, although peptides in both C1 and C2 are pore-formers, they indeed show subtle differences in their physicochemical features when multiple properties are considered jointly (Supplementary Fig. 3).

The two clusters of intracellular-targeting AMPs (C3 and C4) have distinct physicochemical properties. In particular, AMPs in

cluster C4 have an especially high proline content, leading to elevated propensity to intrinsic structural disorder (Fig. 2c), which is a common feature in a novel class of intracellular-targeting AMPs[29]. Indeed, the two AMPs in cluster C4 - Bactenecin 5 (BAC5) and cathelicidin PR-39 – are known to have intracellular targets only as they do not lyse the membrane (Table 1). By contrast, AMPs in cluster C3 show features of both membrane- and intracellular-targeting ones (Fig. 2). Reassuringly, Indolicidin (IND) and Protamine (PROA), which are in cluster C3, have been described to have both membrane disruptive and intracellular-targeting activities (Table 1). Finally, while CAP18 is generally considered as membrane-targeting, our data indicate that it could

also have intracellular targets as it clusters with PROA in the chemical-genetic map (Fig. 2a). Additional work should elucidate the exact mode of action of this peptide.

Taken together, AMPs with similar chemical-genetic interaction profiles share physicochemical features and previously described broad mechanisms of action, indicating that chemical-genetics can capture certain differences in the bactericidal effects across AMPs.

**Functionally diverse genes influence AMP susceptibility.** Functional gene classification revealed that our chemical-genetic hits are involved in diverse biological processes (Supplementary Fig. 4). Importantly, whereas genes annotated with cell envelope function were overrepresented among AMP susceptibility-modulating genes (Supplementary Data 4), the majority of our hits did not have obvious functional connection with known AMP uptake mechanisms or modes of action (Supplementary Fig. 4).

Next, to assess the diversity of resistance determinants across AMPs, we calculated the extent to which the resistance-enhancing genes and the sensitivity-enhancing genes are shared between pairs of AMPs, respectively. To avoid underestimating the overlap between gene sets across AMPs, we employed an index of overlap that takes into account measurement noise (see Methods). Typically, ~63% of the sensitivity-enhancing genes and ~31% of the resistance-enhancing genes overlapped between pairs of AMPs (Supplementary Fig. 5). The latter figure indicates substantial variation in the latent resistome across AMPs. Remarkably, the sets of resistance-enhancing genes varied greatly even between AMPs in the same chemical-genetic cluster, in particular between AMPs in cluster C3 (Fig. 3a). This pattern could reflect subtle differences in the modes of action across the intracellular-targeting AMPs within cluster C3 as these peptides differ in their specific targets (Table 1). Indeed, on a broader scale, membrane-targeting AMP pairs (C1–C2) and intracellular-targeting AMP pairs (C3–C4) shared more resistance-enhancing genes than AMP pairs with different broad mechanisms of action (Fig. 3b). We note that the same conclusions were reached when similarities between AMPs were calculated by correlating their chemical-genetic profiles (see Supplementary Fig. 6a, b).

Finally, we tested whether the above results could be distorted by potential non-specific chemical–genetic interactions induced

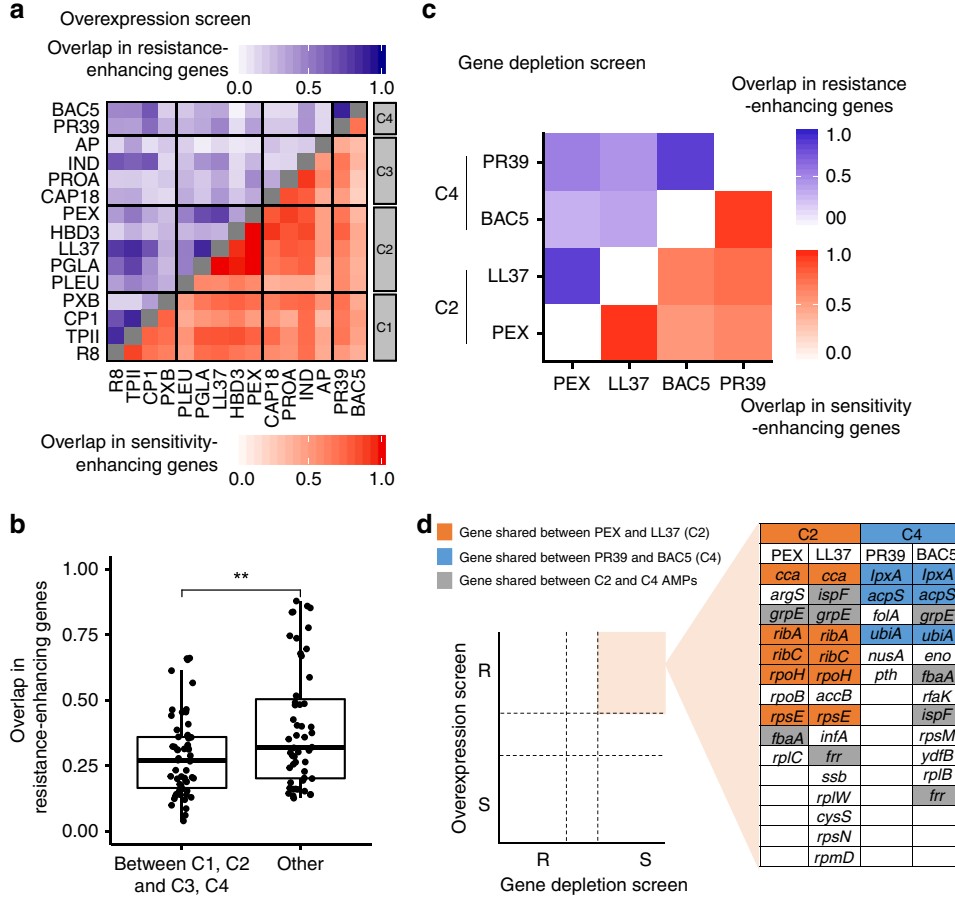

**Fig. 3 Functionally diverse latent and intrinsic AMP resistomes. a** Heatmap shows the corrected Jaccard similarity indices calculated for resistance-enhancing genes (blue) and sensitivity-enhancing genes (red) between AMP pairs based on the overexpression screen (see Methods for calculation of corrected Jaccard indices, n = 210, that is, the number of AMP pairs). The darker the color the higher the overlap of gene sets between AMP pairs. Source data are provided as a Source Data file. **b** The overlaps in the latent resistomes (genes enhancing resistance upon overexpression) between AMP pairs belonging to different chemical-genetic clusters. Significant difference: **P = 0.0045 from two-tailed unpaired t-test, n = 54 and n = 51 for between C1, C2, and C3, C4, and others, respectively. Boxplots show the median (center horizontal line), the first and third quartiles (bottom and top of box, respectively), with whiskers showing either the maximum (minimum) value or 1.5 times the interquartile range of the data. **c** Heatmap shows the corrected Jaccard similarity indices calculated for resistant (blue) and sensitive (red) chemical-genetic interactions with partially depleted essential genes (see Methods). Source data are provided as Supplementary Data 5. **d** Schematic figure showing sets of essential genes that simultaneously enhance AMP resistance when overexpressed and sensitivity when depleted. Color code is explained in the figure.

by the fitness defect of overexpression[30]. To this end, we divided *E. coli* genes into two groups based on the presence or absence of overexpression growth defect[31] and repeated the above analyses on both groups separately (see Methods). Reassuringly, the extent of overlap of chemical-genetic interactions remained highly similar in both groups (Supplementary Fig. 7a-d), indicating that our results are not confounded by overexpression growth defects. In sum, these findings reveal a vast diversity of resistance determinants across peptides that reflects differences in their modes of action and specific targets.

**Depletion of essential genes reveals intrinsic AMP resistance.** Chemical-genetic profiling based on gene depletion captures a different aspect of resistance determinants than gene over-expression[32]. While resistance upon increased gene dosage informs on the latent resistome, hypersensitivity upon gene depletion reveals genes that contribute to resistance at their native expression levels, collectively called as the intrinsic resistome[24]. To investigate the intrinsic AMP resistome, we initiated a chemical-genetic screen with a set of 279 partially depleted essential genes (hypomorphic alleles; see Methods) of *E. coli*. We selected four AMPs with well-characterized modes of action, including two exclusively membrane-targeting (Pexiganan (PEX) and LL37 from C2) and two exclusively intracellular-targeting AMPs (BAC5 and PR39 from C4). Then, using a well-established high-density agar plate assay[19,33], we determined their chemical-genetic interaction profiles across the hypomorphic alleles (Supplementary Data 5). Additionally, we also profiled four small-molecule antibiotics with distinct modes of action in order to rule out that the chemical-genetic profiles are dominated by non-specific chemical-genetic interactions arising from general effects associated with gene depletion (Supplementary Data 5).

In total, we found that 75% of the 279 partially depleted essential genes influenced susceptibility to at least one of the AMPs studied and 60% of these interactions caused hypersensitivity, indicating that essential genes often contribute to the intrinsic AMP resistome (Supplementary Data 5). We found substantial overlaps in the intrinsic resistomes between AMPs with similar modes of action. As high as 87% of the 279 hypomorphic alleles overlapped between PEX and LL37, and a similar figure emerged from the comparison of the gene set between BAC5 and PR39 (Fig. 3c). In contrast, we observed a significantly lower, on average, 59% overlap in intrinsic resistomes between functionally dissimilar AMPs (Fig. 3c, Supplementary Fig. 8). Importantly, the chemical-genetic interactions profiles of AMPs differed markedly from those of antibiotics (Supplementary Fig. 9), indicating that the obtained chemical-genetic interaction profiles are specific to AMPs and not due to general effects associated with the depletion of essential genes.

Genes that simultaneously enhance drug resistance when overexpressed and sensitivity when depleted are of special interest as such genes are likely to directly protect bacteria against drug stress or encode drug targets[34]. Comparison of our overexpression and hypomorphic screens revealed multiple essential genes that showed both properties (Fig. 3d). Remarkably, *folA* (dihydrofolate reductase), a known intracellular target of PR39[35], was among the set of 6 genes that simultaneously conferred resistance when overexpressed and sensitivity when depleted in the presence of PR39. Together, these results indicate that both the intrinsic and the latent AMP resistomes are shaped by the AMP's mode of action.

**Collateral sensitivity between functionally dissimilar AMPs.** The limited overlap in resistance determinants across AMPs prompted us to hypothesize that some of the gene over-expressions might even have antagonistic effects against distinct AMPs. Specifically, we sought to identify resistance-enhancing genes that induce collateral sensitivity, i.e., increase resistance to one AMP while simultaneously sensitize to another one[36,37]. We found numerous such cases (Supplementary Data 6). For example, out of the 4,400 genes, we retrieved 436 that conferred resistance to 2 or more AMPs while increasing sensitivity to at least 2 other AMPs upon overexpression.

For each pair of AMP, we then calculated the overrepresentation of collateral sensitivity-inducing genes over random expectation (see Methods). Intriguingly, pairs of AMPs within the same chemical-genetic cluster were typically depleted in such genes (Fig. 4a). In contrast, the relative overrepresentation of collateral sensitivity-inducing genes was pronounced between the clusters of membrane-targeting and intracellular-targeting AMPs (Fig. 4b). Finally, we observed a similar pattern in the hypomorphic allele screen. Specifically, collateral sensitivity interactions were prevalent between functionally dissimilar AMPs (Supplementary Fig. 10).

**Perturbed phospholipid trafficking induces collateral sensitivity.** We next focused on genes that showed reduced susceptibility to at least four membrane-targeting AMPs (i.e., AMPs from C1 and C2 clusters) while at the same time showed elevated susceptibilities towards at least four intracellular-targeting AMPs (i.e., AMPs from C3 and C4 clusters) upon overexpression. These genes were enriched in functions related to phospholipid and lipopolysaccharide (LPS) composition of the bacterial membrane (Supplementary Fig. 11). This trend is exemplified by MlaD and MlaE proteins (Supplementary Fig. 11a), both being part of a protein complex that carries out retrograde phospholipid transport from the outer membrane to the inner membrane in Gram-negative bacteria[38]. Importantly, several studies have reported a role of the Mla (maintenance of lipid asymmetry) pathway in bacterial pathogenesis, virulence and antibiotic resistance[39,40].

What could be the mechanism behind the antagonistic action of this pathway on membrane- versus intracellular-targeting AMPs? Since MlaD is part of a protein complex, it may lead to a loss-of-function effect upon overexpression[41,42]. To test this, we asked whether overexpression and deletion of *mlaD* cause similar changes in susceptibility to a representative set of membrane- and intracellular-targeting AMPs. Both mutations caused a decreased susceptibility to membrane-targeting AMPs and an increased susceptibility to intracellular-targeting ones (Fig. 5a, for MIC curves, see Supplementary Figs. 12, 13), demonstrating that overexpression perturbs *mlaD* function similar to a loss-of-function mutation.

It has been observed that *mlaD* deletion alters the membrane composition by leading to the accumulation of phospholipids in the outer leaflet of the bacterial outer membrane[38]. A change in membrane composition can alter the net negative surface charge of the cell[3], which in turn strongly influences AMP susceptibility[1]. Thus, we hypothesized that depletion of functional MlaD decreases susceptibility to membrane-targeting AMPs by decreasing the net negative surface charge of the cell. On the other hand, membrane properties can also have an effect on membrane potential[43]. As the uptake of certain intracellular-targeting AMPs, for example, PROA and IND, are driven by membrane potential[44,45], we posited that such an effect could underlie the observed collateral sensitivity interactions. To test this, we measured the net negative surface charge and the membrane potential of the *mlaD* overexpression and deletion strains (see Methods). Reassuringly, both overexpressing and deleting *mlaD* resulted in a significantly decreased negative surface charge

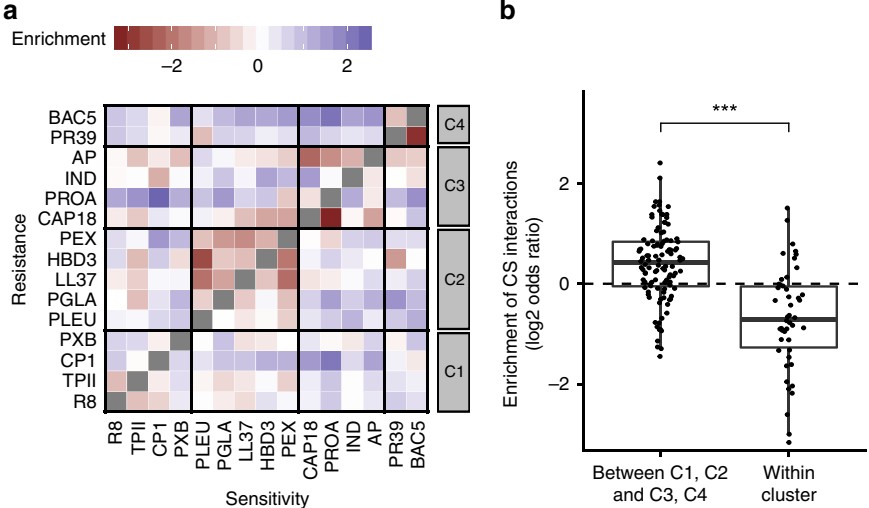

**Fig. 4 Collateral sensitivity (CS) interactions are frequent between AMPs with different modes of action. a** Heatmap depicting the overrepresentation of collateral sensitivity-enhancing genes for each AMP pair over random expectation ($n = 210$ AMP pairs). Random expectation is calculated using the number of resistance-enhancing genes and sensitivity-enhancing genes for each AMP (see Methods). **b** Collateral sensitivity effects were especially pronounced between AMP pairs with different broad mode of action, that is, between membrane-targeting (C1, C2) and intracellular-targeting (C3, C4), as compared to AMP pairs from the same cluster. Significant difference: ***$P = 1.7 \times 10^{-08}$ from two-tailed unpaired $t$-test, $n = 108$ and 46 for pairs of AMPs between C1, C2, and C3, C4, and those within cluster, respectively. $Y$-axis shows odds ratio (log2) of enrichment of collateral sensitivity interactions between AMP pairs. Boxplots show the median (center horizontal line), the first and third quartiles (bottom and top of box, respectively), with whiskers showing either the maximum (minimum) value or 1.5 times the interquartile range of the data. Source data are provided as a Source Data file.

(Fig. 5b) and an increased membrane potential (Fig. 5c and Supplementary Fig. 16).

Finally, we tested whether such correlated changes in surface charge and membrane potential could generally explain antagonistic mutational effects against membrane-targeting and intracellular-targeting AMPs. We therefore investigated three additional randomly selected overexpression strains showing a reduced susceptibility to membrane-targeting AMPs and an elevated sensitivity towards intracellular-targeting AMPs (Supplementary Fig. 14b-d). Consistent with our results on MlaD, all three strains showed a decreased net negative surface charge and an increased membrane potential (Supplementary Fig. 14e, f). As these overexpressed genes represent various biological functions unrelated to phospholipid transfer, these results suggest that perturbed membrane potential and surface charge might be causally involved in the observed collateral sensitivity interactions.

**Limited cross-resistance between functionally dissimilar AMPs.** It has recently been shown that similarities in modes of action and chemical-genetic interaction profiles between antibiotics correlate with the emergence of cross-resistance during laboratory evolution[10,17]. Here we extend this notion to AMPs and hypothesize that AMP pairs with distinct modes of action and chemical-genetic clusters show limited cross-resistance following evolution.

To test this hypothesis, we took advantage of a recent study that (i) generated 38 *E. coli* lines that acquired resistance to one of 8 AMPs through adaptive laboratory evolution (representing both membrane-targeting and intracellular-targeting AMPs from C2 to C4) and (ii) measured susceptibility of these evolved lines relative to that of the parental strain (i.e., relative MIC changes) against a set of 7 AMPs from clusters C1, C2 and C3[20]. Here, we extended this dataset by measuring susceptibility changes of the same evolved lines to four additional AMPs using identical protocols to represent AMPs from all four clusters (Supplementary Fig. 15a, see also Methods). Overall, the resulting dataset of susceptibility

profiles provides a comprehensive map of cross-resistance and collateral sensitivity between AMPs representing various modes of action.

Consistent with the hypothesis, no cross-resistance interaction above a 2-fold MIC increase was observed between exclusively membrane-targeting and exclusively intracellular-targeting AMPs, while cross-resistance was prevalent (~30%) and significantly enriched within both groups (Fig. 6a and Supplementary Fig. 15a). As a further support, collateral sensitivity interactions (i.e., defined as ≥ 20% decrease in MIC) were ~6-fold overrepresented between the groups of membrane-targeting and intracellular-targeting AMPs (Fig. 6b). Importantly, the chemical-genetic clustering provided additional insights into the cross-resistance patterns that could not have been predicted based on the broad mode of action of AMPs. Specifically, while lines adapted to membrane-targeting AMPs from cluster C2 showed widespread cross-resistance to other AMPs from the same cluster, they rarely had such evolutionary interactions with membrane-targeting AMPs from cluster C1 (Supplementary Fig. 15c). Similarly, lines adapted to intracellular-targeting AMPs from cluster C4 showed cross-resistance disproportionally more frequently to C4 AMPs than to intracellularly-targeting AMPs from cluster C3 (Supplementary Fig. 15b). Overall, these findings indicate that chemical-genetic interaction profiles capture genuine differences in resistance mechanisms between AMPs with the same broad mode of action. Finally, we note that AMPs in cluster C3 did not show cross-resistance to each other, confirming their diverse nature (i.e., both intracellular- and membrane-targeting).

## Discussion

This work systematically mapped the genetic determinants of AMP resistance by chemical-genetic profiling in a laboratory strain of *E. coli* (Fig. 1). We report that AMP resistance is influenced, albeit mildly, by a large set of functionally diverse genes, and yet these genes overlap only to a limited extent between AMPs. Specifically, clustering of the chemical-genetic interaction profiles revealed that the modes of action of the AMPs

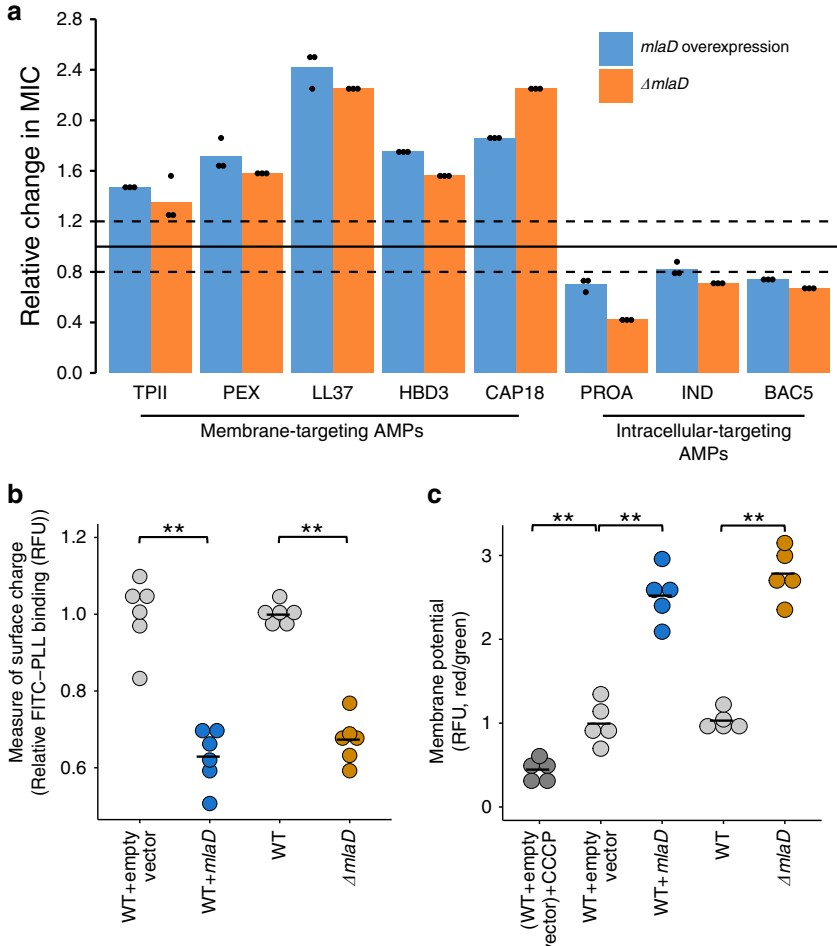

**Fig. 5 Mutation in *mlaD* influences AMP susceptibilities through antagonistic mutational effects. a** Relative change in MICs of the *mlaD* overexpression and deletion strains (*ΔmlaD*) to a representative set of membrane-targeting and intracellular-targeting AMPs. MICs were compared to corresponding wild-type control strains (see Supplementary Figures 12, 13). Dashed lines represent previously defined cut-offs for resistance ($\geq$ 1.2 x MIC of the control) and sensitivity ($\leq$ 0.8 x MIC of the control)[25]. **b** Decreased net negative surface charge of the *mlaD* overexpression and deletion strains. Significant differences: **$P = 0.0021$ and $P = 0.0021$ for WT + empty vector vs overexpression and WT vs deletion strain, respectively, from two-sided Mann–Whitney U test, $n = 6$ biological replicates for each genotype. Charge measurement was done using FITC-labeled poly-L-lysine (FITC-PLL) assay where the fluorescence signal is proportional to the binding of the FITC-PLL molecules. A lower binding of FITC-PLL indicates a less net negative surface charge of the outer bacterial membrane (see Methods). **c** Increased membrane potentials of the *mlaD* overexpression and deletion strains. Significant differences: **$P = 0.007$, $P = 0.0079$ and $P = 0.0079$ for WT + empty vector CCCP control vs WT + empty vector, WT + empty vector vs WT + *mlaD* overexpression and WT vs. *ΔmlaD*, respectively, two-sided Mann–Whitney U test, $n = 5$ biological replicates for each genotype. Relative membrane potential was measured by determining relative fluorescence (RFU) using a carbocyanine dye DiOC2(3) assay (see Methods). Red/green ratios were calculated using population mean fluorescence intensities. WT *E. coli* carrying the empty vector treated with cyanide-m-chlorophenylhydrazone (CCCP, a chemical inhibitor of proton motive force) was used as an experimental control for diminished membrane potential. Central horizontal lines represent mean values of biological replicates. Source data are provided as a Source Data file and in Supplementary Fig. 16.

largely define the gene sets that influence bacterial susceptibility against them (Figs. 2 and 3). Additionally, antagonistic mutational effects are frequent between AMPs that disrupt the bacterial membrane versus those that act on intracellular targets (Figs. 4 and 5). Finally, by capitalizing on a comprehensive set of laboratory-evolved AMP-resistant *E. coli* lineages, we show that cross-resistance rarely occurs between AMPs that belong to distinct modes of action or distinct chemical-genetic clusters (Fig. 6).

The results presented in this study may have important implications for the development of AMP-based therapies. Previous works reported several instances of cross-resistance interactions between membrane-targeting peptides (Supplementary Data 1), however, the potential for cross-resistance across AMPs with different modes of action has remained poorly understood. Specifically, while cross-resistance between host and therapeutic

AMPs is certainly a realistic danger, not all AMPs are equally prone to cross-resistance. Given the immense diversity of AMPs with major differences in physicochemical properties and resistance mechanisms, we propose that carefully chosen therapeutic candidates could mitigate the risk of cross-resistance with specific human host-defense peptides. From our screen, proline-rich AMPs are the best candidates in this respect, supporting the considerable effort that has already been taken into the clinical development of proline-rich AMP-based therapeutic applications[46,47]. Additionally, a distinct group of membrane-targeting AMPs (R8, TPII, and CP1) appear to be less prone to cross-resistance to the investigated human host-defense AMPs. Remarkably, these three AMPs from cluster C1 (Fig. 2a) were the only AMPs in a previous laboratory evolution experiment that did not result in any significant resistant lines[20], further

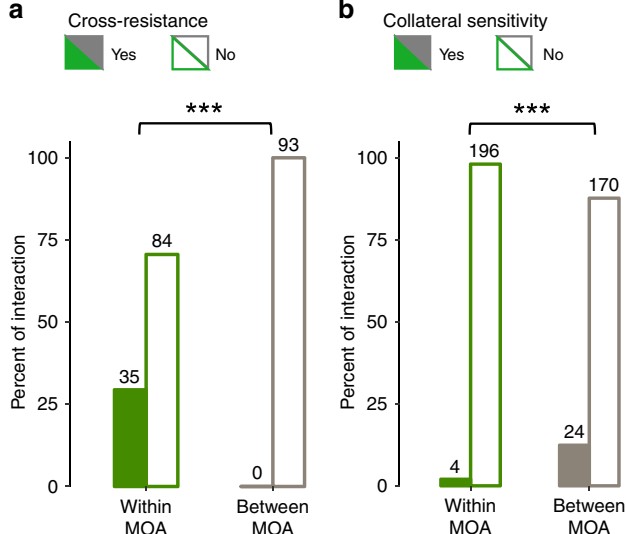

**Fig. 6 Mode of action informs on cross-resistance spectra of AMP-evolved lines. a** Cross-resistance interactions (i.e., defined as 2-fold increase in MIC) are significantly overrepresented between AMP pairs either from the group of exclusively membrane-targeting AMPs (TPII, CP1, PGLA, LL37, PEX) or from the group of exclusively intracellular-targeting AMPs (PR39 and BAC5) as compared to pairs of AMPs between the two mode of action groups (MOA). Significant difference: ***$P = 1.333 \times 10^{-10}$ from two-sided Fisher's exact test, $n = 119$ and 93 for within and between mode of action groups, respectively. **b** Collateral-sensitivity interactions (i.e., defined as $\geq 20\%$ decrease in MIC) are overrepresented between the groups of membrane-targeting and intracellular-targeting AMPs (i.e., between MOA). Significant difference: ***$P = 4.83 \times 10^{-5}$ from two-sided Fisher's exact test, $n = 200$ and 194 for within MOA and between MOA, respectively. Source data are provided as a Source Data file.

corroborating their distinct resistance behavior. Clearly, this work made the first step in this direction and future studies should explore these possibilities. Specifically, cross-resistance patterns of proline-rich AMPs in human saliva and synthetic AMPs should also be considered[48]. Despite these potential therapeutic implications, an important open issue is whether the cross-resistance patterns reported here can be recapitulated in species other than a laboratory *E. coli* strain.

The large diversity of genes that influence AMP resistance upon overexpression indicates that bacterial susceptibility to AMPs is coupled to the general physiology of the bacterial cell, and in particular to alterations in membrane composition. This idea also provides an explanation to a recent finding that antibiotic resistance mutations in membrane proteins frequently induce collateral sensitivity to AMPs through pleiotropic side effects that alter membrane composition[25]. Indeed, the over-representation of collateral sensitivity interactions among AMP resistance determinants implies that evolving AMP resistance requires the optimization of many traits simultaneously. As a consequence, bacterial cells potentially harbor a large mutational target to alter AMP resistance, however, such mutations often have negative trade-offs with other cellular traits.

Collateral sensitivity between AMPs is best exemplified by the Mla pathway. Several studies have reported the importance of Mla pathway in bacterial pathogenesis and virulence[39,40]. For example, loss-of-function mutations in Mla pathway in *Haemophilus influenzae* increased the accumulation of phospholipids in the outer membrane, which mediated sensitivity to human serum[39]. Here, we demonstrated that depletion of *mlaD* decreases the net negative surface charge of the bacterial membrane and,

eventually, causes a somewhat increased resistance to human membrane-targeting AMPs (Fig. 5a, b), and an elevated susceptibility to intracellular-targeting AMPs (Fig. 5a). Together, our work indicates that a trade-off between membrane surface charge and membrane potential underlie collateral sensitivity interactions between membrane-targeting and intracellular-targeting AMPs upon perturbing the Mla pathway. We speculate that this trade-off could contribute to the observed variation in the expression level of Mla pathway proteins among clinical isolates of *H. influenzae*[39].

Whereas the mutations identified in the chemical-genetic screen generally provided relatively small increases or decreases in AMP susceptibilities, these small changes may have clinical implications for several reasons. First, mutations causing low levels of antibiotic resistance may ensure bacterial survival in antibiotic-treated hosts, as it was shown in *Pseudomonas aeruginosa* isolates from cystic fibrosis patients[49]. Second, multiple small-effect resistance mutations, which typically emerge at low antimicrobial concentrations, may combine to confer clinically relevant high-level of resistance[50]. Third, weak collateral sensitivity effects of antibiotic resistance mutations substantially increased the killing efficacy of AMPs against multidrug-resistant bacteria[25].

Our results also have implications for an important but unresolved issue: why have natural AMPs that are part of the human innate immune system remained effective for millions of years without detectable resistance in several bacterial species? One possibility, supported by our work, is that bacteria may have difficulty to evolve resistance to the combination of multiple defense peptides deployed by the immune system due to negative trade-offs between them. We do not claim, however, that AMPs in clinical use would generally be resistance-free. Rather, these properties of the AMPs could be beneficial for the development of combination therapies involving AMPs in combination with antibiotics and human host peptides.

## Methods

**Media, bacterial strains and antimicrobial peptides**. Experiments with AMPs were conducted in minimal salts (MS) medium supplemented with $MgSO_4$ (0.1 mM), $FeCl_3$ (0.54 μg per ml), thiamin (1 μg per ml), casamino acids (0.2%) and glucose (0.2%). Luria-Bertani (LB) medium contained tryptone (0.1%), yeast extract (0.05%), and NaCl (0.05%). All components were purchased from Sigma-Aldrich. To increase the dosage of each *Escherichia coli* gene for the chemical-genetic screen, we used the *E. coli* K-12 Open Reading Frame Archive library (ASKA)[21] in *Escherichia coli* K12 BW25113 cells. AMPs were custom synthesized by ProteoGenix, except for Protamine and Polymyxin B, which were purchased from Sigma-Aldrich. AMP solutions were prepared in sterile water and stored at −80 °C until further use.

**Plasmid DNA preparation and purification**. Bacterial cells harboring the ASKA plasmids were grown overnight in LB medium supplemented with chloramphenicol (20 μg per ml). Cells were harvested by centrifugation. Plasmid DNA isolation was performed using innuPREP plasmid mini Kit (Analytik Jena AG) according to the manufacturer's instructions. To remove the genomic DNA contamination, the isolated plasmid DNA samples were digested overnight with Lambda exonuclease and exonuclease I (Fermentas) at 37 °C. The digested plasmid DNA samples were purified with DNA Clean & Concentrator™ (Zymo) kit according to the manufacturer's instructions.

**Chemical-genetic profiling**. We carried out chemical-genetic profiling to determine the impact of the overexpression of each *E. coli* ORF on bacterial susceptibility to each of the 15 different AMPs. To this end, we used the complete set of *E. coli* K-12 Open Reading Frame Archive (ASKA) plasmid library (GFP minus) where each *E. coli* ORF is cloned into a high copy number expression plasmid (pCA24N-ORFGFP(-)). Prior to screening, the ASKA library was grown in the original host strain *E. coli* K-12 AG1 in 96-well plates (growth conditions: 37 °C, 280 rpm, LB medium). An equal aliquot of each member of the ASKA library (each well of the 96-well plates) was pooled together and the plasmid DNA (pCA24N-ORF-GFP(-)) was isolated and transformed into *E. coli* K12 BW25113 strain[51]. To obtain a negative control strain not having any overexpressed gene, the plasmid without a cloned ORF (pCA24N-noORF) was also

transformed into the same *E. coli* strain. Then, on the pooled collection, we applied a previously reported competitive growth assay[25]. Specifically, the pooled overexpression library and the control strain were grown in parallel in MS medium supplemented with 20 µg per ml chloramphenicol and the overexpression was induced by 100 µM isopropyl-ß-D-thiogalactopyranoside (IPTG). After 1 h induction, ~5 × 10$^5$ bacterial cells from the library were inoculated into each well of a 96-well microtiter plate containing a concentration gradient of an AMP in the MS medium supplemented with 20 µg per ml chloramphenicol and 100 µM IPTG. At the same time, both the library and the control strain with the empty plasmid were grown in the absence of any AMPs. We took special care to grow both of these samples in the exact same conditions as the samples in the presence of AMPs. Bacterial growth was monitored in a microplate reader (Biotek Synergy 2) for 24 h. At the end of the exponential growth phase, we selected those wells in which the doubling time of the cell population was increased by 2-fold. Then, from these wells, cells were split into four equal proportion and each was transferred into 20 mL of MS medium supplemented with the corresponding AMP in four different concentrations in the range that slowed down growth by two-fold in the microtitre plate. Then, following exponential growth, out of the four 20 mL cultures those that showed again a two-fold increase in doubling time were selected for further analysis. The rationale for this 2-step process was to maintain competition in exponential phase for 12 generations of growth, efficiently control the growth rate in a reproducible manner and obtain the plasmid pool with standard DNA isolation protocol (innuPREP plasmid mini Kit, Analytik Jena AG) in a yield that is enough for the downstream analysis. The cultures were vigorously shaken along the entire protocol to make sure that the cultures are completely homogeneous and therefore biasing interactions between clones (such as trans-resistance) was minimized. Each of the selected plasmid samples was digested overnight with a mixture of lambda exonuclease and exonuclease I (Fermentas) at 37 °C to remove the genomic DNA background. The digested plasmid DNA samples were purified with DNA Clean & Concentrator™ (Zymo) kit according to the manufacturer's instructions. This protocol was carried out in two biological replicates for each AMP treatment. In the case of the untreated sample (in the absence of AMP), we had five replicates. *E. coli* BW25113 strain carrying the empty vector was used as a negative control to measure read counts originating from genomic DNA contamination during plasmid preparation (background).

**Deep sequencing of plasmid pool**. The cleaned plasmid samples were sequenced with the SOLiD next-generation sequencing system (Life Technologies) and the relative abundance of each plasmid was determined, as described previously[25,51]. Briefly, the isolated plasmid pool samples were fragmented and subjected to library preparation. Library preparation and sequencing was performed using the dedicated kits and the SOLiD4 sequencer (Life Technologies), respectively. For each sample, 20–25 million of 50 nucleotide long reads were generated. Primary data analysis was carried out with software provided by the supplier (base-calling). The 50 nucleotide long reads were analyzed, quality values for each nucleotide were determined using the CLC Bio Genomics Workbench 4.6 program.

**Data analysis of chemical-genetic screen**. Raw sequence data processing and mapping onto *E. coli* ORFs were carried out as described previously[25]. Raw sequence data were also mapped to the plasmid backbone. In order to make the mapped read counts comparable between the different samples, we carried out the following data processing workflow based on established protocols[52,53], using a custom-made R script. The extra read counts deriving from genomic DNA contamination (background) were estimated by assuming that the reads mapping to the unit length of the plasmid and the ORFs should have a ratio of 1:1. The total extra read count estimated thereof was partitioned among the ORFs based on their background frequency (that is, their relative frequency obtained from the experiment involving the empty plasmid). Next, these ORF-specific backgrounds were subtracted from the read counts. Then, a loglinear transformation was carried out on the background-corrected relative read counts. Compared to the canonical logarithmic transformation, this transformation has the advantage of avoiding the inflation of data variance for ORFs with very low read counts[54]. The transformed relative read counts showed bimodal distributions (Supplementary Fig. 17). The lower mode of the distribution corresponds to ORFs that were not present in the sample. The upper mode represents those ORFs whose growth was unaffected by overexpression (i.e., no fitness effect). To make different samples comparable, the two modes of the distribution of each sample were set to two predefined values. These values were chosen such that the original scale of the data was retained. In order to align the modes between samples, we introduced two normalization steps: one before and one after loglinear transformation. The first normalization step identified the lower mode corresponding to the absent strains and added a constant to shift the lower mode to zero. Next, we performed the loglinear transformation step described above. The second normalization step was a linear transformation moving the upper mode to a higher predefined value. Following these normalization steps, genes that were close to the lower mode in the untreated samples were discarded from the analysis as these represent strains that displayed poor growth even in the absence of drug treatment (that is, AMP sensitivity could not be reliably detected). A differential growth score (i.e., fold-change) was calculated for each gene as the ratio of the normalized relative read

counts in treated and non-treated samples at the end of the competition. Fold-change values of biological replicate experiments were averaged. To determine fold-changes that are statistically significant, we estimated the variance of biological replicate measurements as follows. Due to small sample sizes ($n = 2$ for AMP treatments and $n = 5$ for untreated competitions), gene-specific variance estimates are unreliable. Therefore, we shared information across multiple genes and AMP treatments by calculating the standard deviation as a function of normalized read counts using lowess smoothing (i.e., local regression). This procedure is based on the observation that the variance depends on the mean. Note that similar strategies are commonly used in the gene expression literature[55]. Using this estimation of standard deviation, we applied z-tests to determine whether the treated and the non-treated samples differ significantly. Genes that showed at least 2-fold lower and higher relative abundance with a $p$-value $< 0.05$ at the end of the competition upon AMP treatment were considered as sensitizing and resistance-enhancing genes, respectively.

**Cluster analysis of chemical-genetic interaction profiles**. To group AMPs with similar chemical-genetic interaction profiles, we employed an ensemble clustering algorithm that combines multiple clustering results to obtain a robust clustering[56]. A combination of diverse clustering results based on perturbing the input data and clustering parameters is known to yield a more robust grouping of data points than that obtained from a single clustering result.

As a first step, we removed genes that did not show AMP-specific phenotypes across treatments since these genes would be uninformative for clustering. To this end, we retained only those genes that showed significant differences in their fold-change values between AMPs compared to their variances across replicate measurements within AMPs as assessed by F-tests ($p < 0.01$). This resulted in a set of 2146 genes kept for clustering. Next, we employed a distance metric, normalized variation of information, to measure distances between AMP chemical-genetic interaction profiles. The normalized variation of information is closely related to mutual information but has the advantage of being a true distance metric. Importantly, normalized variation of information gives more weight to rare overlaps of resistance/sensitivity phenotypes between AMPs, unlike the commonly used Euclidean distance. Normalized variation of information ($NVI$) between AMP pairs was calculated as follows: $NVI = (H - I) / H$ where $H$ is the entropy and $I$ is the mutual information.

Based on this distance measure, we then generated 75,000 clusters of AMPs by perturbing both the AMP profile data and the clustering parameters. The AMP profile data was perturbed by resampling the gene set with replacement and by randomly selecting a single chemical-genetic interaction profile among the multiple biological replicates available for each AMP. We used hierarchical clustering and varied both the algorithms (Ward, single-linkage, complete-linkage and average-linkage) and the number of clusters defined ($k = 2...6$). Results of the 75,000 clusters were summarized in a consensus, which contains, for each pair of AMP, the number of times that two AMPs cluster together across all of the clustering results. Finally, we clustered this consensus matrix using hierarchical clustering and complete linkage and plotted the result as a heatmap.

**Construction of hypomorphic alleles for chemical-genetic screening**. A total of 279 essential gene hypomorphs (with reduced protein expression) were constructed essentially, as previously described[18,19]. Briefly, as with the mRNA perturbation by DAmP (decreased abundance by mRNA perturbation) alleles in yeast[57], we created an essential gene hypomorphic mutation by introducing a kanamycin (Kan$^R$) marked C-terminal sequential peptide affinity fusion tag, engineered by homologous recombination into each essential gene[58]. The tag perturbs the 3′ end of the expressed mRNA of the essential proteins, when combined with environmental/chemical stressors, or other mutations by destabilizing the transcript abundance. A subset of these hypomorphic alleles that we used[19,59] or shared with others[14] have revealed functionally informative gene-gene, and gene-environment or drug–gene interactions.

Analogous to our *E. coli* synthetic genetic array approach[59], our chemical-genetics screening strategy involves robotic pinning of each Kan$^R$ marked single essential gene hypomorph arrayed in 384 colony format on Luria Broth (LB) medium, in quadruplicate, onto the minimal medium containing AMPs under a selected concentration, in two replicates, generating eight replicates in total for each essential gene hypomorph. The sub-inhibitory concentration was chosen based on 50% growth inhibition of wild-type cells using a serial dilution. In parallel, we also prepared two replicates of control plates containing arrayed essential gene hypomorphic strains pinned onto minimal media without AMPs. After incubation at 32 °C for 20 h, the plates (with and without AMPs) were digitally imaged and colony sizes were extracted from the imaged plates using an adapted version of the gitter toolbox[60]. The resulting raw colony size (proxy for cell growth) from each screen, with and without AMP, was normalized using SGAtools suite[61], with default parameters. The normalized colony sizes from the AMP plate was subtracted from their corresponding colony screened without AMP to estimate the final hypomorphic-strain fitness score (sensitive or resistant), which is as an average of all eight replicate measurements recorded for each hypomorphic allele. A z-score distribution based $p$-value was calculated for all interactions and those with $p \leq 0.05$ were deemed as significant interactions. To group the chemical-

genetic interaction profiles of AMPs and antibiotics, we employed hierarchical clustering on Euclidean distances with Ward's method.

**Physicochemical properties of AMPs**. Protein amino acid frequencies were counted with an in-house perl script. Isoelectric point, hydrophobicity, hydrophobic moment, net charge and membrane surface was calculated with the peptides R package, version 2.4[62]. The ExPasy Prot Param tool was used for calculating molecular weight and peptide length[63].

**Differentiation between AMP clusters based on physicochemical parameters**. Logistic regression framework was used with two parameters to infer differences between C1 and C2 clusters in the peptides physicochemical properties. Area under the receiver operating characteristic curve (ROC) was used to establish model accuracies and rank parameter pairs using the caret (v6.0–80) and e1071 (v1.7–0) R packages. For a global analysis of cluster properties, principal component analysis was applied to all the peptide physicochemical properties with centering and scaling the data using the princomp R package. All calculations were done in R version 3.5.0 in Rstudio version 1.1.447[64,65].

**Calculating the overlap in resistance-enhancing and sensitivity-enhancing gene sets between AMPs**. To calculate the extent to which the resistance and sensitivity-enhancing genes are shared between pairs of AMPs, we used a modified version of the Jaccard index that takes into account measurement noise. Specifically, for each pair of AMP, we calculated the Jaccard index of overlap between their sets of resistance-enhancing genes and performed a correction by dividing this value by the average Jaccard index of overlap between replicate screens of the same AMPs. Thus, a corrected Jaccard index value of 1 between two AMPs indicates that the set of resistance-enhancing genes overlap as much as that of two replicate screens. Protein overexpressions associated with a fitness cost were identified using a previously published dataset[31]. In this dataset, those overexpression strains were deemed as having a fitness cost where the doubling time was at least two-fold higher than the doubling time of the strain harboring the no-insert control of the pCA24N vector.

**Enrichment of collateral sensitivity interactions between AMP pairs**. We calculated the overrepresentation of collateral sensitivity-inducing genes for each AMP pair over random expectation using data from our overexpression screen. Random expectation was calculated using the number of resistance-enhancing and sensitivity-enhancing genes for each AMP. Enrichment ratio ($r$) of collateral sensitivity-inducing genes for each AMP pair was calculated as follows:

$$r = x/e \qquad (1)$$

where: $x$ - actual frequency of the genes showing collateral sensitivity interactions between AMP pair $e$ - expected frequency (based on marginal probability) of the genes showing collateral sensitivity interactions between AMP pair. Expected frequency ($e$) was calculated as follows:

$$e = R_{amp1} * S_{amp2} \qquad (2)$$

where: $R_{amp1}$ = relative frequency of genes showing resistance to AMP1 out of all ~4400 genes screened $S_{amp2}$ = relative frequency of genes showing sensitivity to AMP2 out of all the ~4400 genes screened.

**Gene ontology (GO) enrichment analysis**. To determine which Gene ontology (GO) terms are significantly enriched in the resistance-enhancing and sensitivity-enhancing genes, we employed the Biological Networks Gene Ontology tool (BiNGO)[66]. The selection of GO reference set was based on the EcoGene database[67]. The Benjamini-Hochberg FDR (FDR cutoff = 0.05) was used for multiple-testing correction[68]. GO categories showing FDR-corrected $P$-values < 0.05 were considered statistically significant. Detailed information about the significantly enriched GO categories is provided in Supplementary Data 4.

We calculated the enrichment of phospholipid and lipopolysaccharide transport/binding functions among the genes that showed collateral sensitivity. Genes related to phospholipid and lipopolysaccharide transport/binding function were selected from a previous study[25].

**Determination of minimum inhibitory concentration (MIC)**. Minimum inhibitory concentrations (MIC) were determined with a standard serial broth dilution technique with a minor modification[69]. Specifically, instead of a two-fold dilution series, smaller AMP concentration steps were used (typically 1.2-fold) for two reasons. First, AMPs have steeper dose-response curves than standard antibiotics, and therefore two-fold dilutions were not enough to capture 90% growth inhibitions (i.e., MIC). Second, gene overexpressions caused typically only small changes in the MICs of AMPs and therefore higher resolution was required. The protocol was as follows. From a stock solution of an AMP, 12-steps serial dilution was prepared in fresh MS medium in 96-well microtiter plates. Each AMP was represented in 11 different concentrations (3 wells per AMP concentration per strain). Three wells contained only medium to check the growth in the absence of AMP. After overnight growth in MS medium supplemented with chloramphenicol,

bacterial strains were diluted 20-fold into fresh MS medium and grown until the cell density reached $OD_{600}$ ~1. Cells were induced by 100 μM of IPTG and incubated for 1 h at 30 °C with continuous shaking at 300 rpm. Following incubation, approximately half-million cells were inoculated into the wells of the 96-well microtiter plate with a 96-pin replicator. We used three independent replicates for each strain and the corresponding control. Two rows in the 96-well plate contained only MS medium in order to obtain the background OD value of the medium. Plates were incubated at 30 °C with continuous shaking at 300 rpm. After 20–24 h of incubation, $OD_{600}$ values were measured in a microplate reader (Biotek Synergy 2). After background subtraction, MIC was determined as the lowest concentration of AMP where the $OD_{600}$ values were less than 0.05.

**Membrane surface charge measurement**. To evaluate bacterial surface charge, we performed a fluorescein isothiocyanate-labeled poly-L-lysine (FITC-PLL) (Sigma) binding assay. In brief, FITC-PLL is a polycationic molecule that binds to anionic lipid membrane in a charge-dependent manner and is used to investigate the interaction between cationic peptides and charged lipid bilayer membranes[70]. The assay was performed as previously described[25,71]. Briefly, bacterial cells were grown overnight in MS medium, centrifuged and washed twice with 1X PBS buffer (pH 7.4). The washed bacterial cells were re-suspended in 1× PBS buffer to a final $OD_{600}$ of 0.1. A freshly prepared FITC-PLL solution was added to the bacterial suspension at a final concentration of 6.5 μg/ml. The suspension was incubated at room temperature for 10 min, and pelleted by centrifugation. The remaining amount of FITC-PLL in the supernatant was measured fluorometrically (excitation at 500 nm and emission at 530 nm) with or without bacterial exposure. The quantity of bound molecules was calculated from the difference between these values. A lower binding of FITC-PLL indicates a less net negative surface charge of the outer bacterial membrane.

**Membrane potential measurement**. A previously described protocol[36] was used to determine the change in transmembrane potential for *mlaD* overexpression and *mlaD* knockout strains in comparison to their control strain. Transmembrane potential (Δψ) was measured using the BacLight™ Bacterial Membrane Potential Kit (Invitrogen). In this assay, a fluorescent membrane potential indicator dye emits green fluorescence in all bacterial cells and the emission shifts to red in the cells that maintain a high membrane potential. In this way, the ratio of red/green fluorescence provides a measure of membrane potential. Prior to the measurement bacterial cells were grown overnight in MS medium at 30 °C. The overnight cultures were diluted into fresh MS medium and grown until cell density reached $OD_{600}$ 0.5–0.6. The grown cultures were diluted to $10^6$ cells/mL in filtered PBS buffer. Then, 5 μl of 3 mM $DiOC_2(3)$ was added to each sample tube containing 500 μl of bacterial suspension and incubated for 20 min at room temperature. Following incubation, red to green fluorescence values of the samples were measured using Fluorescence Activated Cell Sorter (BD Facscalibur) according to the instructions of the kit's manufacturer. Fluorescence values were calculated relative to the control strain. Control populations treated with cyanide-m-chlorophenylhydrazone (CCCP, a chemical inhibitor of proton motive force) were used as an experimental control.

**MlaD knockout construction**. A *mlaD* knockout strain of *E. coli* BW25113 carrying a kanamycin resistance cassette in the position of the gene was selected from the KEIO collection[72]. The resistance marker was removed using plasmid-borne (pFT-A) expression of FLP recombinase leading to excision of the kanamycin resistance cassette[73]. Cassette excision was verified by a polymerase chain reaction using the primers mlaD_del_ver_Fw (5′- TCACGGTGACGTGGATTTC) and mlaD_del_ver_Rev (5′- GCCTCGTCCATCAGCTTATAC).

**Cross-resistance interactions of AMP-resistant lineages**. To characterize the cross-resistance of the 38 *E. coli* lineages we obtained relative MIC changes that are, the MIC values of the evolved lines compared to these of the parental strain, from two data sources. First, we used already measured relative MIC changes from Spohn et al. Supplementary data 4[20]. This data covers the cross-resistance to seven AMPs from cluster C1 (TPII, CP1 and R8), C2 (LL37, PLEU, PGLA) and C3 (IND). Second, we extended this dataset by measuring the relative MIC changes of the same evolved lines to four additional AMPs using identical protocol[20]. The later data set covered cross-resistance to AMPs from cluster C2 (PEX), C3 (PROA) and C4 (PR-39 and BAC5). In this way, AMPs from each cluster were represented in the dataset.

**Reporting summary**. Further information on research design is available in the Nature Research Reporting Summary linked to this article.

## Data availability

All data generated or analyzed during this study are present in this article and its Supplementary Information files. For each figure, the availability of the analyzed data is indicated in the figure legend. The source data underlying Figs. 3a, b, 4a, b, 5a-c and 6 and Supplementary Figs 1, 5, 6, 7, 12, 13, 14e, f and 15 are provided as a Source Data file. The SOLiD sequencing data for the chemical-genetic screen is available in the NCBI Sequence Read Archive, with SRA accession number PRJNA576179. Any additional data can be requested from the corresponding author.

## Code availability

All scripts and other files needed to reproduce the chemical-genetic interaction score calculations and data analyses are available at https://github.com/pappb/Kintses-et-al-AMP-Chemogenomics-NatComm.

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

## Acknowledgements
We thank Roland Tengölics for helpful discussions. This work was supported by the 'Lendület' programme of the Hungarian Academy of Sciences (B.P. and C.P.), the Wellcome Trust (B.P.), The European Research Council H2020-ERC-2014-CoG 648364-Resistance Evolution (C.P.), GINOP- 2.3.2–15–2016–00014 (EVOMER, C.P. and B.P.), GINOP-2.3.2–15–2016–00020 (MolMedEx TUMORDNS, C.P.) and GINOP-2.3.2–15–2016–00026 (iChamber, B.P.), National Research, Development and Innovation Office, Hungary NKFIH grant K120220 (B.K.), NKFIH grant FK124254 (O.M.) and NKFIH grant KH125616 (B.P.), Natural Sciences and Engineering Research Discovery Grant 20234 and Canadian Institutes of Health Research (CIHR) project grant 148831 (M.B) and The European Union's Horizon 2020 research and innovation programme under grant agreement No 739593 (B.P. and B.K.). A.Ga. is supported by a CIHR Postdoctoral Fellowship. B.K. holds a Janos Bolyai Research Fellowship from the Hungarian Academy of Sciences and is supported by the UNKP-18-4 New National Excellence Program of the Ministry of Human Capacities. O. M. holds a Janos Bolyai Research Fellowship from the Hungarian Academy of Sciences.

## Author contributions
B.K., C.P., and B.P. conceived the project. B.K., P.K.J., G.F., C.P., and B.P. planned experiments and data analyses. P.K.J. and M.S. performed most experiments. R.S., L.D., A.M., and V.L. carried out cross-resistance measurements. I.N. was responsible to SOLiD sequencing. B.C. carried out mutagenesis. B.K., P.K.J., G.F., O.M., and E.A. analyzed experimental data. P.K.J., G.F., and A.G. carried out bioinformatic analyses. A.H., A.Ga., and S.K. created all essential hypomorphic alleles, and performed the chemical-genetic screening. S.P quantified the colony growth fitness of the hypomorphs, and analyzed the data with input from M.B. B.K., P.K.J., C.P., and B.P. wrote the manuscript with input from all co-authors.

## Competing interests
I.N. had consulting positions at SeqOmics Biotechnology Ltd. at the time the study was conceived. SeqOmics Biotechnology Ltd. was not directly involved in the design and execution of the experiments or in the writing of the manuscript. This does not alter the author's adherence to sharing data and materials. The rest of the authors declare no competing interests.

## Additional information

**Peer Review Information** *Nature Communications* thanks the anonymous reviewers for their contribution to the peer review of this work. Peer reviewer reports are available.

