## [Peer Review File · Nature Communications]

Reviewers' comments:

Reviewer #1 (Remarks to the Author):

The authors have addressed my original queries on their manuscript. I have no further issues.

Reviewer #2 (Remarks to the Author):

The authors have answered my comments from the previous review.

My outstanding concern relates to the how the insights gained will benefit study of AMPs therapeutically as the authors indicate. The average increase in MIC to gene overexpression is reported as ~1.6-fold. It is unclear how a 1.6-fold effect would impact consideration for AMP development since MIC₉₀ measurements frequently range an order of magnitude while still receiving clinical use. In this context, overexpression of the same resistance genes in a second WT (non-lab) *E. coli* may not show these small effects. Is there an example where small changes like these in sensitivity have impacted the use or development of clinical drugs?

Reviewer #3 (Remarks to the Author):

The authors substantially revised the manuscript; in particular, they de-emphasized the collateral sensitivity aspects and emphasized that AMPs with different modes of action rarely show cross-resistance. I think that this shifted emphasis fits much better with the novel aspects of this work. As

a result, the revised manuscript is considerably improved and likely suitable for publication in Nature Communications. A few points still need to be addressed:

Major points

1. Cross-resistance spectra: Figure 6 shows that the mode of action of AMPs can partially predict cross-resistance. However, these data do not directly show that chemical-genetic interaction profiles predict cross-resistance spectra. Also, it seems that Figure 6a shows essentially the same data as Figure 3 from the related manuscript (which is also under consideration in Nature Communications). It seems redundant to show these data in a main figure in both papers.

2. Identification of relevant genes: Throughout the manuscript, there are claims about the high numbers of genes identified as, for example, resistance-enhancing genes. These claims are based on the supposed 'high sensitivity' (line 128) of the method applied. However, while I am convinced that chemical-genetic profiling is a powerful tool to detect gene clusters, it appears less reliable as a tool for identifying individual genes. For example, while the Pearson correlation coefficient of 0.63 reported in Figure 1b shows that replicate experiments are moderately correlated, it also shows that there is considerable variability at the single-gene level. Therefore, any claims about numbers of genes or specific genes being identified are doubtful (unless the specific genes are validated in independent experiments). A thorough statistical analysis (estimating false positive rates etc.) could help to improve the estimates about numbers of genes. Alternatively, these claims could simply be removed and the authors could instead focus on clusters of larger numbers of genes (and correlations between them) – which is anyway what most of the paper does.

3. Use of thresholds in the data analysis: Currently, similarities between AMPs are identified by counting how many resistance-enhancing genes overlap. This digitized approach unnecessarily discards a lot of quantitative information in the data, namely the level of resistance enhancement. Why not go for a more nuanced approach: For example, normalize the fold-change increase in the presence of an AMP by the fold-change increase in the absence of AMP (solving the problem stated in line 186-187), and then correlate this normalized fold-change increase between drugs using a Pearson correlation coefficient. This approach uses more of the data and is not sensitive to the choice of threshold for defining a gene as 'resistance-enhancing'.

Minor points

1. Line 121: “are not necessarily expected to alter...in the expected direction” – rephrase into ‘do not necessarily alter’.

2. Line 209-211: duplicate text (same as 206-208) should be removed.

3. Line 336-337: the ‘previous works’ should be referenced.

Point-by-point responses to Referee's comments

Reviewer #1 (Remarks to the Author):

The authors have addressed my original queries on their manuscript. I have no further issues.

Thank you.

Reviewer #2 (Remarks to the Author):

The authors have answered my comments from the previous review. My outstanding concern relates to the how the insights gained will benefit study of AMPs therapeutically as the authors indicate. The average increase in MIC to gene overexpression is reported as ~1.6-fold. It is unclear how a 1.6-fold effect would impact consideration for AMP development since MIC90 measurements frequently range an order of magnitude while still receiving clinical use. In this context, overexpression of the same resistance genes in a second WT (non-lab) *E. coli* may not show these small effects. Is there an example where small changes like these in sensitivity have impacted the use or development of clinical drugs?

We thank the Referee for the comment. We made several changes to the text to address this issue.

First, we now emphasize better that most of the susceptibility changes are of small effect. For example, the second sentence of the discussion is now as follows:

“We report that AMP resistance is influenced, albeit mildly, by a large set of functionally diverse genes” (line 322)

Second, we have also introduced a new paragraph into the discussion to address the question on the clinical relevance of resistance mutations with small effects.

“Whereas the mutations identified in the chemical-genetic screen generally provided relatively small increases or decreases in AMP susceptibilities, these small changes may have clinical implications for several reasons. First, mutations causing low levels of antibiotic resistance may ensure bacterial survival in antibiotic-treated hosts, as it was shown in *Pseudomonas aeruginosa* isolates from cystic fibrosis patients (Frimodt-Møller, J. *et al. Sci. Rep.* 2018). Second, multiple small-effect resistance mutations, which typically emerge at low antimicrobial concentrations, may combine to confer high levels of resistance (Wistrand-Yuen, E. *et al. Nat. Commun.* 2018). Third, weak collateral sensitivity effects of antibiotic resistance mutations substantially increased the killing efficacy of AMPs against multidrug-resistant bacteria (Lázár, V. *et al. Nat. Microbiol.* 2018).” (lines 371-379)

Finally, the text now better emphasizes that all experiments were carried out in a laboratory strain of the model bacterium *E. coli* and the direct applicability of our findings to pathogenic bacteria remains to be seen. In particular, we have added this sentence to the discussion:

“Despite these potential therapeutic implications, an important open issue is whether the cross-resistance patterns reported here can be recapitulated in species other than a laboratory *E. coli* strain.” (lines 347-349)

Reviewer #3 (Remarks to the Author):

The authors substantially revised the manuscript; in particular, they de-emphasized the collateral sensitivity aspects and emphasized that AMPs with different modes of action rarely show cross-resistance. I think that this shifted emphasis fits much better with the novel aspects of this work. As a result, the revised manuscript is considerably improved and likely suitable for publication in Nature Communications. A few points still need to be addressed:

Thank you.

Major points

1. Cross-resistance spectra: Figure 6 shows that the mode of action of AMPs can partially predict cross-resistance. However, these data do not directly show that chemical-genetic interaction profiles predict cross-resistance spectra. Also, it seems that Figure 6a shows essentially the same data as Figure 3 from the related manuscript (which is also under consideration in Nature Communications). It seems redundant to show these data in a main figure in both papers.

As the Referee suggested we moved Figure 6a into the supplementary.

2. Identification of relevant genes: Throughout the manuscript, there are claims about the high numbers of genes identified as, for example, resistance-enhancing genes. These claims are based on the supposed 'high sensitivity' (line 128) of the method applied. However, while I am convinced that chemical-genetic profiling is a powerful tool to detect gene clusters, it appears less reliable as a tool for identifying individual genes. For example, while the Pearson correlation coefficient of 0.63 reported in Figure 1b shows that replicate experiments are moderately correlated, it also shows that there is considerable variability at the single-gene level. Therefore, any claims about numbers of genes or specific genes being identified are doubtful (unless the specific genes are validated in independent experiments). A thorough statistical analysis (estimating false positive rates etc.) could help to improve the estimates about numbers of genes. Alternatively, these claims could simply be removed and the authors could instead focus on clusters of larger numbers of genes (and correlations between them) – which is anyway what most of the paper does.

To address this, we removed all claims about the number of AMP susceptibility modulating genes from the manuscript. We retained only those specific claims on individual genes that were validated by independent experiments.

In line 125 'high sensitivity' was removed.

3. Use of thresholds in the data analysis: Currently, similarities between AMPs are identified by counting how many resistance-enhancing genes overlap. This digitized approach unnecessarily discards a lot of quantitative information in the data, namely the level of resistance enhancement. Why not go for a more nuanced approach: For example, normalize the fold-change increase in the presence of an AMP by the fold-change increase in the absence of AMP (solving the problem stated in line 186-187), and then correlate this normalized fold-change increase between drugs using a Pearson correlation coefficient. This approach uses more of the data and is not sensitive to the choice of threshold for defining a gene as 'resistance-enhancing'.

Regarding point 3, we carried out the suggested analysis with Pearson correlations. Here we wish to emphasize that we slightly modified the standard approach as a simple Pearson correlation of the fold-change values cannot handle the different degrees of overlap of resistance and sensitive genes (i.e. both positive and negative values contribute to the correlation coefficient). Because we were interested in whether AMPs show different levels of similarities for resistant and sensitive chemical-genetic interactions, we calculated two separate Pearson correlation coefficients by converting all fold-change values below our significance threshold to zero (i.e. to measure similarity of resistance interactions, we

converted all fold changes <2 and $P > 0.05$ in the profiles to zero and then calculated the correlation coefficient between these modified profiles). This procedure also has the advantage that it minimizes the influence of noisy data points that do not correspond to real chemical-genetic interactions. Note that quantitative information is still retained as the amount of fold-change above the threshold is kept. Reassuringly, this new analysis based on Pearson correlations fully confirmed our previous conclusions obtained by calculating overlaps between resistance / sensitive gene sets. However, the new analysis didn't give any additional insight and therefore we include it as a new supplementary figure only (Supplementary Figure 6).

Of note, we have already been using fold-change values that are normalized to competition in the absence of AMP, as suggested by the reviewer.

Minor points

1. Line 121: "are not necessarily expected to alter...in the expected direction" – rephrase into 'do not necessarily alter'.

Done

2. Line 209-211: duplicate text (same as 206-208) should be removed.

Done

3. Line 336-337: the 'previous works' should be referenced.

Done

REVIEWERS' COMMENTS:

Reviewer #3 (Remarks to the Author):

The authors have addressed all remaining comments. I support publication without any need of further modifications.